# Self-Supervised Diffusion Model Sampling with Reinforcement Learning

## Abstract

Diffusion models have established themselves as the state-of-the-art for generative modeling, dethroning Generative Adversarial Networks (GANs) by generating higher-quality samples while remaining more stable throughout training. However, diffusion models generate samples iteratively and remain slow at inference time. Our work proposes to leverage reinforcement learning (RL) to accelerate inference by building on the recent framing of diffusion's iterative denoising process as a sequential decision-making problem. Specifically, our approach learns a scheduler policy that maximizes sample quality while remaining within a fixed budget of denoising steps. Importantly, our method is agnostic to the underlying diffusion model and does not re-train it. Finally, unlike previous RL approaches that rely on supervised pairs of noise and corresponding denoised images, our method is self-supervised and directly maximizes similarity in dataset feature space. Overall, our approach offers a more flexible and efficient framework for improving diffusion model's inference in terms of speed and quality.

## 1 Introduction

For the past decade, the trend of generative modeling was dominated by Generative Adversarial Networks (GANs) (Goodfellow et al., 2014). While they were considered the state of the art and enjoyed blazing fast inference speeds, they suffered from major training instabilities, namely mode collapse and sensitivity to hyper-parameters (Arjovsky & Bottou, 2017; Wiatrak et al., 2020). These shortcomings have encouraged the search for a more reliable generative modeling paradigm, and has given rise to to the emergence of diffusion models (Sohl-Dickstein et al., 2015; Song & Ermon, 2020a;b). While both GANs and diffusion models are predicated on transforming random noise (generally a standard Gaussian) to match a data distribution, the generator of GANs attempts to do so in a single network pass, while in diffusion models, this transformation is sequential, allowing to trade off sample quality for inference speed. While there has been work to speed up diffusion models by retraining them with different objective functions (Song et al., 2023; Heek et al., 2024), we propose a simpler approach that does not modify or retrain the underlying diffusion model, but rather learns an optimal sampler through the use of reinforcement learning (RL).

We frame the diffusion model sampling process as an RL episode, where each inference pass is an episode step. We treat each element in a batch of data as its own RL agent, allowing for a paralellized environment, where batches of trajectories can be collected from a single diffusion sampling pass. This lends itself perfectly for online reinforcement learning algorithms such as Proximal Policy Optimization (PPO) (Schulman et al., 2017), allowing for an extremely fast flow of data into the RL learner.

## 2 Background

In this section, we provide a background on the topics at hand, namely RL and Markov decision processes (MDPs), diffusion models, and learned denoise schedulers.

## 2.1 RL & MDPs

An MDP is a framework to formalize sequential decision-making problems. It is defined by a state space $\mathcal{S}$, which is the set of possible states, the action space $\mathcal{A}$, which is the set of possible actions, the transition probabilities $P$, where $P : \mathcal{S} \times \mathcal{A} \times \mathcal{S} \to [0, 1]$ is the probabilty of transitioning from a given state $s_k$ to a new state $s_{k+1}$ when taking action $a_k$, formally given by $P(s_{k+1}|s_k, a_k)$, and finally the reward function $R$, where $R : \mathcal{S} \times \mathcal{A} \to \mathbb{R}$ is the immediate reward $r_{k+1}$ observed when taking action $a_k$ at state $s_k$, formally given by $R(s_k, a_k)$. Together, the MDP is defined by the tuple $\mathcal{M} = \{\mathcal{S}, \mathcal{A}, P, R\}$. (Sutton & Barto, 2018; Kaelbling et al., 1996)

Given a trajectory of state and actions $\tau = (s_0, a_0, s_1, a_1, ..., s_K, a_K)$, the goal of an RL agent is to maximize the expected cumulative reward over the entire trajectory. This expectation is with respect to its policy $\pi(a|s)$, which is a function that returns a probability distribution over all actions, given a state $s$: $\mathbb{E}_\pi[\sum_{k=0}^{K} R(s_k, a_k)]$. To the more seasoned RL researcher, the timestep of the MDP is usually denoted by $t$ rather than $k$. However, we reserve that notation to diffusion models, as they also express their framework in the time domain, both continuous and discrete.

## 2.2 DIFFUSION MODELS

Diffusion models are a type of generative model known for their ability to generate high-quality samples from a given dataset. While various different formulations of diffusion models have emerged over the last few years (Song et al., 2022; Ho et al., 2020; Song et al., 2021; Heitz et al., 2023), all proposed methods tackle the same problem of sampling from a complex distribution via a learned transformation from a simpler distribution, usually Gaussian.

While this idea isn't new (Rezende & Mohamed, 2016; Kingma & Welling, 2022; Goodfellow et al., 2014), none of the previous methods were able to achieve a quality of samples comparable to diffusion models, which is mainly attributed to their iterative inference architecture. To transform sampled Gaussian noise into a sample that attempts to match the dataset distribution, multiple sequential denoising steps must be applied. This paradigm differentiates diffusion models from its predecesors, where the learned transformation was a single function evaluation.

Given samples from a data distribution $\mathbf{x}_0 \sim q(\mathbf{x}_0)$, diffusion models are tasked to learn $p_\theta(\mathbf{x}_0)$ which approximates $q(\mathbf{x}_0)$: $p_\theta(\mathbf{x}_0) = \int p_\theta(\mathbf{x}_{0:T})d\mathbf{x}_{1:T}$. Here, the joint distribution $p_\theta(\mathbf{x}_{0:T})$ is referred to as the reverse process. It is a series of learned transformations, with an initial fixed starting point of $p(\mathbf{x}_T) = \mathcal{N}(\mathbf{x_T}; 0, \mathbf{I})$. Intuitively, these are a Markov Chain of denoising operators on an initial purely Gaussian noise, which is trivial to generate samples from, by modeling $p_\theta(\mathbf{x}_{0:T}) = p(\mathbf{x}_T) \prod_{t=1}^{T} p_\theta(\mathbf{x}_{t-1}|\mathbf{x}_t)$. The choice of operator varies from different diffusion model formulations. The most common (Ho et al., 2020) transitions are learned isotropic Gaussians, with $p_\theta(\mathbf{x}_{t-1}|\mathbf{x}_t) = \mathcal{N}(\mathbf{x}_{t-1}; \mu_\theta(\mathbf{x_t}, \mathbf{t}), \sigma_t^2\mathbf{I})$, where $\mu_\theta(\mathbf{x}_t, t)$ is the learned mean of the reverse transition. In order to have ground truth information to train our reverse process, the forward process $q(\mathbf{x}_{1:T}|\mathbf{x}_0)$ is a Markov Chain of fixed Gaussian noise which gradually adds noise to the data, with $q(\mathbf{x}_{1:T}|\mathbf{x}_0) = \prod_{t=1}^{T} q(\mathbf{x}_t|\mathbf{x}_{t-1})$. Each transition in the forward process is fixed, and follows a variance schedule $\beta_1, ..., \beta_T$, formally defined as $q(\mathbf{x}_t|\mathbf{x}_{t-1}) = \mathcal{N}(\mathbf{x}_t; \sqrt{1 - \beta_t}\mathbf{x}_{t-1}, \beta_t\mathbf{I})$. In this formulation, $\beta_t = \sigma_t^2$.

Once the diffusion is trained to approximate the target distribution, we can then start generating data from random noise. The quality of the final sample is influenced by two key factors: the numbers of steps T the model takes to denoise the sample, and the noise schedule across those T steps, which controls how the denoising process is distributed. While it may seem that there is a monotonic relationship between the number of inference steps T and the quality of samples, this is not always the case. In fact, it will depend on the training dynamics of the diffusion model. Likewise, the choice of optimal noise schedule is not trivial, and will vary from different diffusion models.

## 3 RELATED WORK

The sensitivity of sample quality to the denoising schedule has given rise to extensive research efforts aimed at identifying and developing optimal scheduling strategies.

Two main school of thoughts have emerged to tackle this problem, the first one are considered **training-free**, which are not reliant on any learning based algorithm, but rather analytically solving for the noise schedule which theoretically guarantees convergence (Lu et al., 2022; 2023), reducing discretization (Zhang & Chen, 2023; Zheng et al., 2023c), higher order solvers (Dockhorn et al., 2022), treating Diffusion Models as manifolds and applying pseudo-numerical methods (Liu et al., 2022). While all of these methods can achieve state of the art results with impressive speed ups, they often rely on hand-crafted heuristics and parameters.

The second school of thought, which are **training-based**, are further split into two camps: methods that rely on training entirely new diffusion models, by either learning the optimal transition operator (Zheng et al., 2023a), truncating the diffusion process by learning different initial noise representations (Zheng et al., 2023b), or approaches akin to knowledge distillation (Song et al., 2023; Heek et al., 2024). Other training-based methods treat the diffusion model as a black box, and either learn model and dataset specific denoising schedules by minimizing the Kullbcak-Leibler Upper Bound (KLUB) between the true reverse-time SDE integration and its time discretization (Sabour et al., 2024), Or use Reinforcement Learning to align the diffusion process of a denoising scheduler with a larger number of steps to that of a model with fewer steps, ensuring the results remain consistent. (Wang et al., 2023). The primary drawback of this approach is that the teacher schedule is expensive to execute and acts as a ceiling to the quality of the generated samples. Furthermore, determining the optimal number of steps to achieve high-quality samples for guidance is not trivial, even under the assumption of unlimited budget, as additional steps don't always result in better quality.

## 4 METHODOLOGY

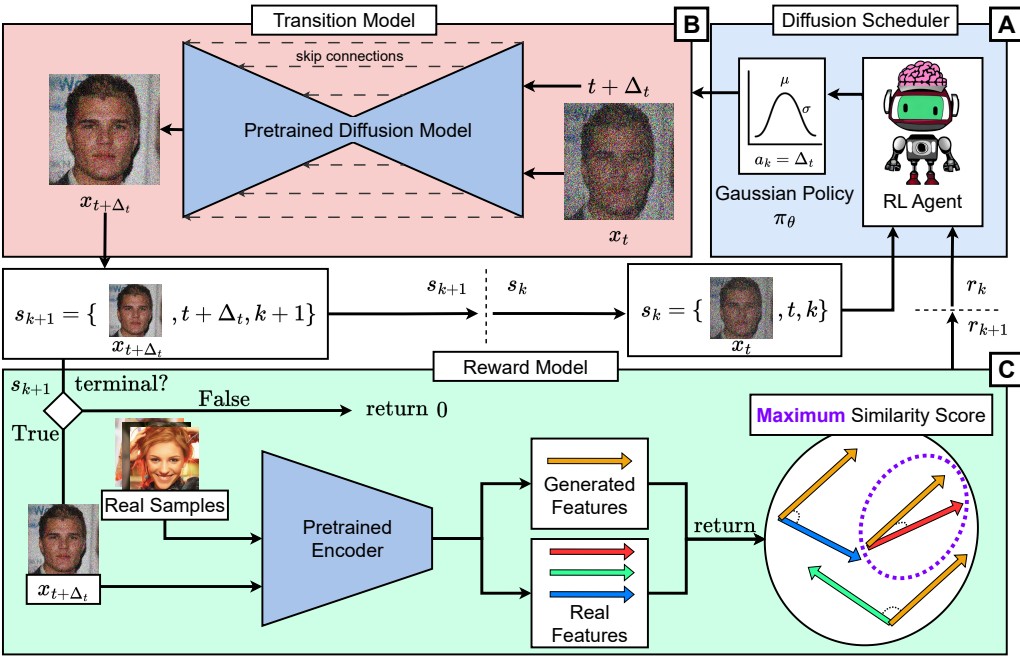

Figure 1: **RL Environment.** The reinforcement learning agent observes the current diffusion sample and noise schedule, from which it decides how to conduct the noise schedule update through its action. It receives a reward based on the maximum pairwise similarity score on extracted features between the generated sample and positive samples drawn form the target dataset.

The main objective of our method is to accelerate the sampling process of a pretrained diffusion model, using reinforcement learning. The goal is to have a lightweight module that would learn to maximize sample quality, by optimizing the noise scheduling, for a given number of maximum inference steps T. Unlike traditional schedulers, our method is allowed to terminate before reaching its budget.

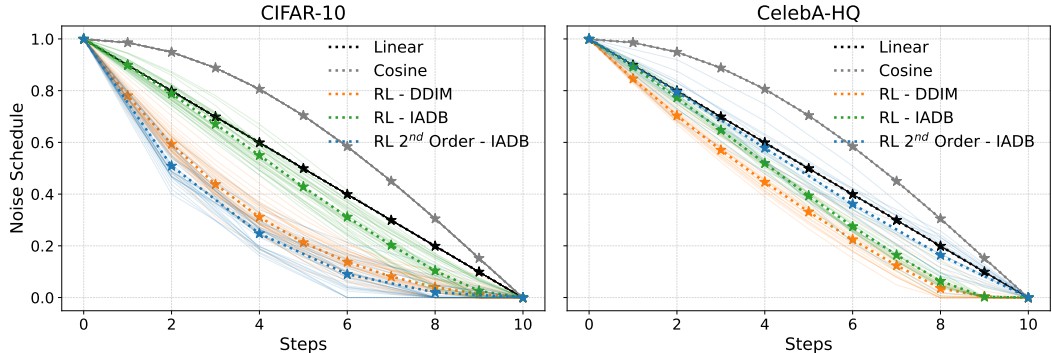

Figure 2: **Comparing Noise Schedules.** We compare the RL schedule to linear and cosine schedules, on a batch of 64 samples for. For both CIFAR-10 and CelebA-HQ, we showcase the RL DDIM model, RL IADB model, and RL IADB second order model. The RL schedule is designed to individually control and adapt the noise schedule of each sample of the batch individually.

Importantly, our agents are **self-supervised**, meaning we do not rely on teacher schedules or paired data, allowing for fast and unbouded sample quality. In the sections below, we break down the formulation of the RL environment into its Markov Decision Process (MDP) components, that is: the state-space, action-space, transition dynamics, and reward function.

## 4.1 REINFORCEMENT LEARNING ENVIRONMENT

We formulate the diffusion sampling process as a reinforcement learning episode. The **state** $s_k$ that the RL agent receives from the environment is the *current diffused data sample* $x_t$, along with the *noise schedule*: the current diffusion step $t$, and the *RL time-step*: the current episode step $k$. Initially, at $s_{k=0}$, the diffused data sample is a pure Gaussian noise, along with the initial noise schedule $T$ of the diffusion model, and timestep 0. $s_k = [x_t : \mathcal{N}(0, \mathbf{I}), t : T, k : 0]$

The **action** $a_k$ that the agent can take to act on the environment is the amount of noise update it would like to apply on the current diffusion sample. We rescale all diffusion models to be consistent in terms of noise schedule, and the associated **action space** is $\mathbb{R} \in [0, 1]$. While some diffusion model's formulation is such that a fully noisy sample is at timestep T, and a fully diffused sample is at timestep 0 (Song et al., 2022; Ho et al., 2020), other models represent the noise schedule in the reverse order, with a fully noisy sample being at timestep 0, and an increase in time representing a diffusion (Heitz et al., 2023). In either formulation, the noise schedule can be normalized to be from 0 to 1.

While we do not make the direction of the flow of time consistent between models, this can be easily made consistent post-training. This means, while some diffusion models will have their noise schedule start at 1 and terminate at 0, some will be inverted. A visualization of the noise schedule for a budget of $T = 10$ is shown in figure 2.

Once the action is picked, the environment transition dynamics are simply the underlying diffusion model inference pass, where a single diffusion step is performed, with the requested noise update, and responds to the RL agent with the updated diffusion sample, along with its associated new noise level and an incremented timestep, as well as a **reward** $r(s_k, a_k)$, which is the *similarity score* of the current diffusion sample, more on this in section 4.2.

The episode is considered **terminated** when the updated noise level is equal to, or exceeds the maximum amount permitted, which is 1 for forward flowing models (Heitz et al., 2023), or 0 for backwards flowing models (Song et al., 2022; Ho et al., 2020). We also set a maximum number of allowed diffusion steps in the environment, which the RL agent is made aware of in its state via the timestep to keep the environment Markovian, and terminate the episode if that number is exceeded. This is analogous to giving the agent a certain *budget* that it cannot exceeded in order to terminate its sampling process.

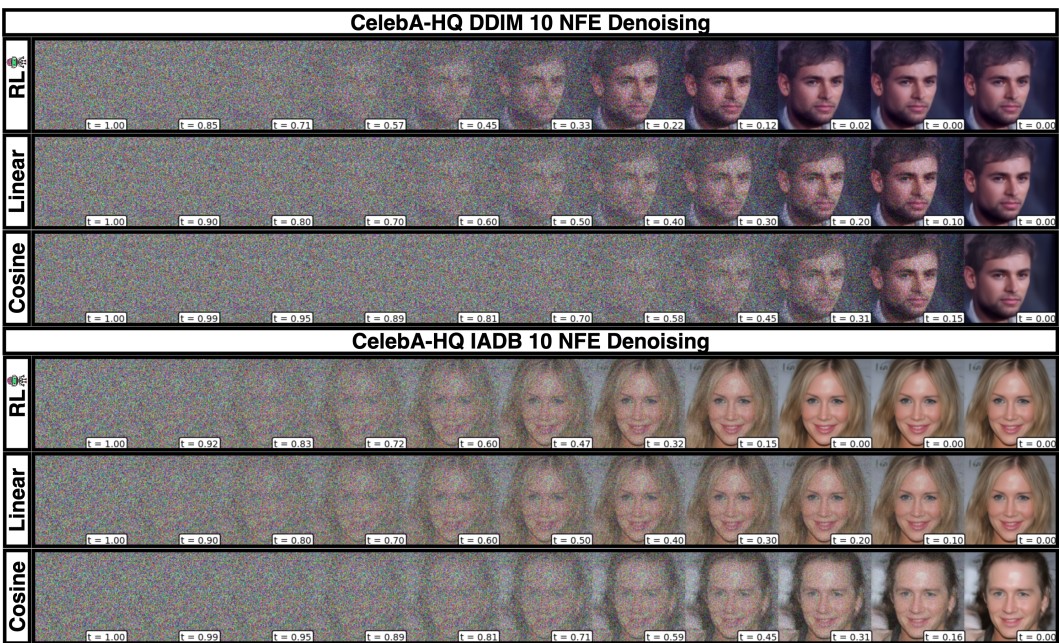

Figure 3: **CelebA-HQ Denoising.** We show the effect of different first order noise schedules for identical initial conditions, for both IADB and DDIM models. The RL scheduler is able to produce higher quality samples with lower inference passes.

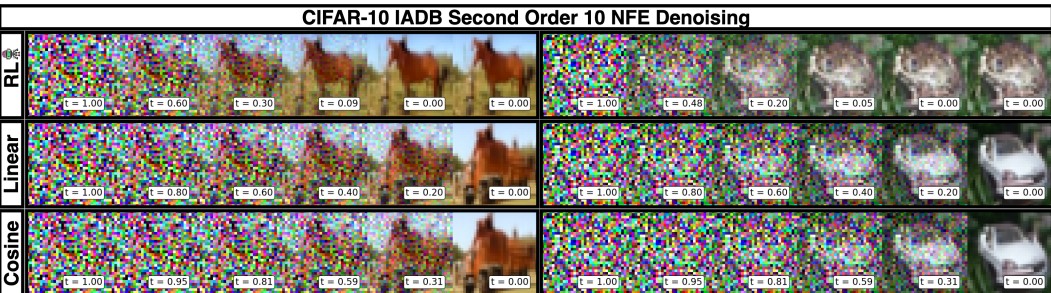

Figure 4: **CIFAR-10 Denoising.** We show the effect of different second order noise schedules for identical initial conditions, for IADB. Despite not producing as high of an FID, the RL scheduler is able to generate sharper images with less class ambiguity.

While traditional reinforcement learning environments are CPU intensive and require explicit parallelization to generate the data required to train, such as OpenAI gym's classical control environments or MuJoCo (Brockman et al., 2016; Todorov et al., 2012), our environment can run entirely on the GPU, as all the dynamics are simulated using a neural network. We can therefor leverage the power of batched computing to parallelize as many environments as our hardware would allow us to. Once we process an entire diffusion pass, where the whole batch is terminated, we then offload this data to the RL learner, keeping each batch element contiguous to preserve the trajectories of our individual agents, where our PPO learner can then proceed with its update.

## 4.2 REWARD PAIRING

At the beginning of each episode, alongside our batch of diffusion trajectories, we sample a batch of ground truth data $D$, which will be used as samples for our reward signal. The reward that the agents observe is sparse, being 0 everywhere, except the final step, when they produce their final diffusion sample. We extract the features of the final samples using a pre-trained Inception-v3 network (Szegedy et al., 2015) from the Pytorch-Lightning library (Paszke et al., 2019; Falcon &

The PyTorch Lightning team, 2019), which is the same model used to compute the Fréchet inception distance (FID) (Heusel et al., 2018). Alongside the diffused sample Inception features, we also extract the features of the samples $D$. We then compute a pairwise similarity matrix between each diffused sample and each example $d_i \in D$.

It is important to emphasize here that we do not have any pre-determined pairing between our generated samples and our ground truth data. Therefor, in order to associate a meaningful reward singal, we extract the maximum similarity value for each diffused sample, which will serve as our final episodic reward. Taking the maximum similarity helps the diffusion model align its sample as best as it can with the highest likelihood data point in the batch. For some datasets with both high *inter* and *intra* class variability (e.g. CIFAR-10) it is a pseudo class-guidance without explicitely giving the labels to our policy.

Having experimented with many different similarity metrics $S_\phi$, we empirically observed that the maximum Pearson Correlation was performing the best. As such, our reward function $R(s_k, a_k)$ can be expressed as:

$$R(s_k, a_k) = \begin{cases} \max_{d_i \in D} S_\phi(x_{t+\Delta_t}, d_i) & \text{if } s_{k+1} \text{ is terminal} \\ 0 & \text{otherwise} \end{cases} \tag{1}$$

where $S_\phi(x, y)$ is defined as:

$$S_\phi(x, y) = \frac{\left(f_\phi(x) - \overline{f_\phi(x)}\right) \cdot \left(f_\phi(y) - \overline{f_\phi(y)}\right)}{\left\|f_\phi(x) - \overline{f_\phi(x)}\right\| \left\|f_\phi(y) - \overline{f_\phi(y)}\right\|}, \quad \overline{f_\phi(x)} = \frac{1}{K} \sum_{k=1}^{K} f_\phi(x)_k \tag{2}$$

where $f_\phi$ is our pretrained feature extractor, and $\overline{f_\phi(x)}$ denotes the mean of the features across the feature dimension $K$ for a given input vector $x$. In our environment, we use the full 2048 features of the Inception-v3 model.

This reward function encourages the agent to produce samples that maximize similarity with the samples $D$, which are sampled from the original dataset. A full depiction of the our method is shown in figure 1. Since we are interested in the absolute highest quality achievable within our budget, we set the discount factor $\gamma = 1.0$. This means, our policy will always aim to maximize the quality of the samples, so long as it stays within budget . In an ideal world with infinite compute power, we would set $D$ to be equal to the entire dataset, and not a sub-sample of it. We show empirically however, that our approach is sound, as well as provide a theoretical grounding to our approach in Appendix A.

### 4.3 POLICY NETWORK ARCHITECTURE

Since our state $s_k$ is a combination of 3-D image data $x_t$ along with floats $t$ and $k$, finding a suitable representation is a challenge. We first extract a latent representation of our image data $x_t$ using a convolution block, a latent representation of our floats $t$ and $k$ using a linear block, and a fused latent representation using a bilinear layer. Finally, the resulting feature vector which is simply the concatenation of all extracted latents, is passed through linear blocks to output parameters to a Gaussian policy $\pi_\theta \sim \mathcal{N}(\mu_\theta, \sigma)$, shown in figure 5. Rather than learning the exploration parameter $\sigma$, which was leading to unstable training dynamics, we opted for a fixed variance schedule, that exponentially decays over the duration of the training.

## 5 EXPERIMENTS

We train our method on two different diffusion models, with varying datasets and sampler orders. The first model is a discrete time diffusion model, which is the Denoising Diffusion Implicit Model (DDIM) (Song et al., 2022). It discritizes the noise schedule $t$ from a number $T$ to 0, where $t = T$ represents the pure noise, and $t = 0$ represents the final denoised sample. We use a pretrain weights available on Hugging Face for the CIFAR-10 dataset (Krizhevsky, 2009) as well as the CelebA-HQ dataset (Liu et al., 2015), with $T = 1000$ for both datasets.

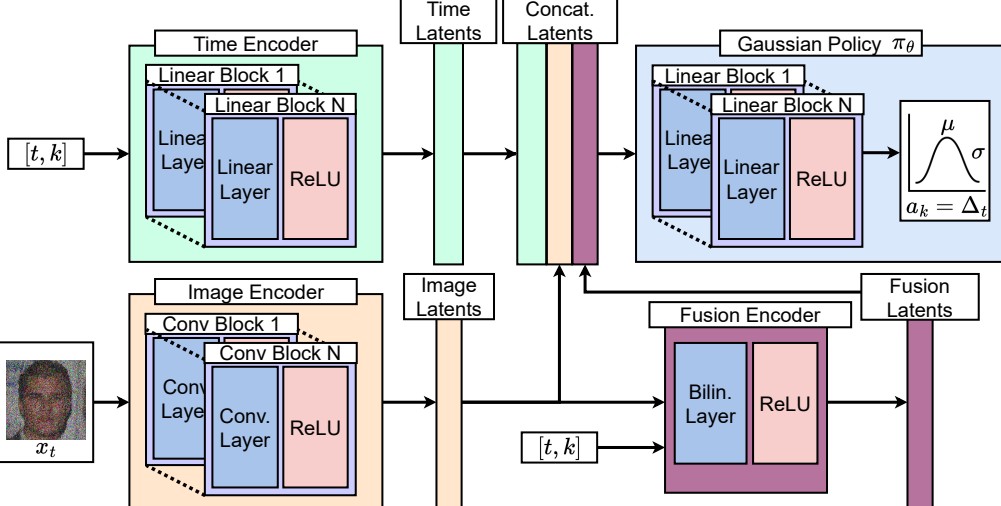

Figure 5: **Policy Network Architecture.** The policy network generates three feature vectors from the inputs. The first is a float latent vector from the timestep $k$ and noise schedule $t$. The second is an image latent vector from the image data $x_t$. The third is a fusion latent vector from the image latent vector and the raw floats, through a bilinear layer. The latents then get concatenated and used as input to a Gaussian policy $\pi_\theta$ which outputs the parameters $\mu$ and $\sigma$ to sample an action $a_k$, which would represent the noise schedule update $\Delta_t$.

The second model is a continuous time diffusion model, which is the Iterative $\alpha$-(de)Blending (IADB) (Heitz et al., 2023). It represents the noise schedule as a continuous number $\alpha$ from 0 to 1, where $\alpha = 0$ represents the pure noise, and $\alpha = 1$ represents the final denoised sample. We train our own diffusion model for CIFAR10 as it is not available, and use a pre-trained model for the CelebAHQ256 dataset, also available on Hugging Face.

As a comparative baseline, we generate FID scores for our DDIM discrete models on traditional first order uniform samplers, as well as the state of the art DPM++ solver (Zheng et al., 2023c; Lu et al., 2022; 2023), straight out of the box from Hugging Face. For our IADB continuous models, we generate FID scores using a uniform and cosine schedule first order solver, as well as a uniform and cosine schedule second order solver. For the implementation of the second order solver, we opt for the Runge-Kutta (RK) midpoint method, as described in Heitz et al. (2023). For each of these baselines, we evaluate them on a varying budget of $T \in [10, 20, 30, 50, 100]$, for a total of 30 different baselines.

Similarly, we train our RL agent to produce a timestep schedule for varying combinations. For DDIM, we train our RL agent on a first order sampler. For IADB, we train our RL agent on both first order and second order samplers. For the second order sampler, the RL agent will still implement a midpoint second order sampler, to ensure fairness with the baselines.

## 6    RESULTS

In this section, we denote our previously mentionned budget $T$ as Neural Function Evaluations (NFE), since some samplers perform multiple diffusion model passes per step. We therefor compare against equal number of diffusion model passes, and not total timesteps. We note that the DPM++ solver was unable to produce meaningful FID scores ($\geq 400$), as such, we do not report the performance of the DPM++ in our results tables. All results are reported in tables 1, 2, 3, 4, 5. While we compare samplers with equivalent NFEs, our RL cannot be directly matched with traditional samplers. While it is trained with a specified NFE budget, it is not forced to use all of it, generating dynamic and adaptive sampling rollouts.

Table 1: FID scores across various datasets, diffusion models, and samplers, for 10 NFE budgets.

| Dataset | Model | FID (↓) on 50k Samples with 10 NFE budget | | |
|---|---|---|---|---|
| | | Uniform Steps | Cosine Steps | RL Steps (Ours) |
| **First Order Sampler** | | | | |
| CIFAR-10 | IADB | 9.74 | 10.35 | **8.61** |
| | DDIM | 15.70 | 64.20 | **11.29** |
| CelebA-HQ | IADB | 72.28 | 47.41 | **37.96** |
| | DDIM | 38.98 | 97.79 | **32.11** |
| **Second Order Sampler** | | | | |
| CIFAR-10 | IADB | **3.95** | 4.57 | 14.32 |
| CelebA-HQ | IADB | 32.24 | 25.06 | **24.23** |

Table 2: FID scores across various datasets, diffusion models, and samplers, for 20 NFE budgets.

| Dataset | Model | FID (↓) on 50k Samples with 20 NFE budget | | |
|---|---|---|---|---|
| | | Uniform Steps | Cosine Steps | RL Steps (Ours) |
| **First Order Sampler** | | | | |
| CIFAR-10 | IADB | 4.22 | 4.74 | **3.96** |
| | DDIM | 8.42 | 67.55 | **4.39** |
| CelebA-HQ | IADB | 32.01 | 21.68 | **19.25** |
| | DDIM | 23.75 | 87.13 | **21.12** |
| **Second Order Sampler** | | | | |
| CIFAR-10 | IADB | **2.16** | 2.55 | 9.17 |
| CelebA-HQ | IADB | 10.43 | 7.61 | **6.52** |

Table 3: FID scores across various datasets, diffusion models, and samplers, for 30 NFE budgets.

| Dataset | Model | FID (↓) on 50k Samples with 30 NFE budget | | |
|---|---|---|---|---|
| | | Uniform Steps | Cosine Steps | RL Steps (Ours) |
| **First Order Sampler** | | | | |
| CIFAR-10 | IADB | 3.15 | 3.30 | **2.95** |
| | DDIM | 6.16 | 70.37 | **3.28** |
| CelebA-HQ | IADB | 16.86 | 12.24 | **9.41** |
| | DDIM | 17.67 | 83.75 | **13.44** |
| **Second Order Sampler** | | | | |
| CIFAR-10 | IADB | **2.05** | 2.06 | 8.19 |
| CelebA-HQ | IADB | 5.65 | 3.96 | **3.25** |

Table 4: FID scores across various datasets, diffusion models, and samplers, for 50 NFE budgets.

| Dataset | Model | FID (↓) on 50k Samples with 50 NFE budget | | |
|---|---|---|---|---|
| | | Uniform Steps | Cosine Steps | RL Steps (Ours) |
| **First Order Sampler** | | | | |
| CIFAR-10 | IADB | 2.58 | 2.40 | **2.28** |
| | DDIM | 4.12 | 78.61 | **2.09** |
| CelebA-HQ | IADB | 7.55 | 5.80 | **5.03** |
| | DDIM | 11.78 | 81.41 | **9.58** |
| **Second Order Sampler** | | | | |
| CIFAR-10 | IADB | 2.26 | **1.97** | 7.88 |
| CelebA-HQ | IADB | 3.79 | **2.87** | 2.96 |

# 7 CONCLUSION

We propose a novel approach to sample diffusion models using RL, without the need of teacher examples, or whitebox access to the model, which renders this method both extremely efficient and simple to use. Our method can theoretically work for any integration problem, and is not limited to

Table 5: FID scores across various datasets, diffusion models, and samplers, for 100 NFE budgets.

| Dataset | Model | FID ($\downarrow$) on 50k Samples with 100 NFE budget | | |
| | | Uniform Steps | Cosine Steps | RL Steps (Ours) |
| **First Order Sampler** | | | | |
| CIFAR-10 | IADB | 2.35 | 2.06 | **1.95** |
| | DDIM | 2.38 | 76.83 | **1.46** |
| CelebA-HQ | IADB | 3.79 | 3.06 | **2.86** |
| | DDIM | 8.25 | 81.12 | **7.32** |
| **Second Order Sampler** | | | | |
| CIFAR-10 | IADB | 2.40 | **2.21** | 2.62 |
| CelebA-HQ | IADB | 2.98 | 2.79 | **2.72** |

diffusion models, so long as there is a way to evaluate the output of that integration. We solve this problem for the diffusion model setting using a stochastic proxy of image quality.

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

# A APPENDIX

## A LOWER BOUNDING THE FRÉCHET INCEPTION DISTANCE DUE TO FINITE SAMPLING FROM A NORMAL DISTRIBUTION

The Fréchet Inception Distance (FID) is widely used to evaluate the quality of generative models by measuring the Wasserstein-2 distance between two multivariate normal distributions. However, when estimating FID using finite samples, there exists an unavoidable error due to finite-sample noise. Here, we derive a lower bound on the expected FID when comparing a normal distribution to its empirical estimate from $n$ samples.

**Assumption and motivation.** Natural images are known to concentrate on a low-dimensional manifold. Although the Inception-Net feature space has dimension $d = 2048$, most of the variance lies in a much smaller number of directions. This motivates a *low-rank, bounded-spectrum assumption* for the covariance of Inception features. Formally, we assume the covariance $\boldsymbol{\Sigma} \in \mathbb{R}^{d \times d}$ of the embedding distribution has effective rank $r \ll d$, with eigenvalues $\lambda_1 \geq \cdots \geq \lambda_r > 0$ on its support and $\lambda_{\max} = \lambda_1$ bounding the variance per direction.

**Setup.** Let $\mathcal{N}(\mu, \boldsymbol{\Sigma})$ denote the true Gaussian approximation to the embedding distribution, with mean $\mu$ and covariance $\boldsymbol{\Sigma}$. Given $n$ i.i.d. samples $x_1, \ldots, x_n$, the empirical mean and covariance are

$$\widehat{\mu} = \tfrac{1}{n} \sum_{t=1}^{n} x_t, \qquad \mathbf{S} = \tfrac{1}{n} \sum_{t=1}^{n} (x_t - \widehat{\mu})(x_t - \widehat{\mu})^\top.$$

The squared $W_2$ distance (Fréchet Inception Distance) is

$$\mathrm{FID}(\mathcal{N}(\mu, \boldsymbol{\Sigma}), \mathcal{N}(\widehat{\mu}, \mathbf{S})) = \|\mu - \widehat{\mu}\|^2 + \mathrm{Tr}\big(\boldsymbol{\Sigma} + \mathbf{S} - 2(\boldsymbol{\Sigma}\mathbf{S})^{1/2}\big).$$

**Bounding the expectation under low-rank structure.** Since $(\boldsymbol{\Sigma}\mathbf{S})^{1/2}$ is positive semidefinite, the cross-term only reduces the trace. Thus,

$$\mathrm{FID} \ \leq\ \|\mu - \widehat{\mu}\|^2 + \mathrm{Tr}(\boldsymbol{\Sigma}) + \mathrm{Tr}(\mathbf{S}).$$

Taking expectations and using $\mathbb{E}[\mathbf{S}] = \boldsymbol{\Sigma}$ and $\mathbb{E}\|\mu - \widehat{\mu}\|^2 = \tfrac{1}{n}\mathrm{Tr}(\boldsymbol{\Sigma})$, we obtain

$$\mathbb{E}[\mathrm{FID}] \ \leq\ \left(2 + \tfrac{1}{n}\right)\mathrm{Tr}(\boldsymbol{\Sigma}).$$

Now, under the low-rank bounded-spectrum assumption,

$$\mathrm{Tr}(\boldsymbol{\Sigma}) = \sum_{i=1}^{r} \lambda_i \ \leq\ r\,\lambda_{\max},$$

so the finite-sample expectation bound becomes

$$\boxed{\ \mathbb{E}[\mathrm{FID}] \ \leq\ r\,\lambda_{\max}\left(2 + \tfrac{1}{n}\right).\ }$$

**Interpretation.** This bound shows that the unavoidable FID error from finite samples scales linearly with (i) the effective rank $r$ of the feature covariance and (ii) the largest variance $\lambda_{\max}$ in the spectrum, with a modest multiplicative factor $(2 + 1/n)$. If the spectrum decays quickly so that $r$ is small and $\lambda_{\max}$ is moderate, the finite-sample noise in FID remains small even when $d$ is large. For example, with $n = 256$, $r = 200$, and $\lambda_{\max} = 0.01$, the bound evaluates to

$$\mathbb{E}[\mathrm{FID}] \ \leq\ 200 \times 0.01 \times 2.0039 \ \approx\ 4,$$

consistent with observed FID fluctuations in practice. Note that $n = 256$ is the number of real image samples we draw in our optimization. The values $r = 200$ and $\lambda_{\max} = 0.01$ are justifiable from basic PCA analysis of the datasets (most of the dataset variance can be controlled by 200 principal components in CIFAR-10 and the other datasets).

