# OpenReview forum: "Self-Supervised Diffusion Model Sampling With Reinforcement Learning"
_ICLR.cc/2026/Conference — Submitted to ICLR 2026_

### Official Review · Reviewer_som6 · 2025-10-27

**Soundness:** 2
**Presentation:** 1
**Contribution:** 2
**Rating:** 0
**Confidence:** 5

**Summary:**

This work introduces a reinforcement learning–based scheduler policy that optimizes sample quality under a fixed number of denoising steps, without retraining the diffusion model. The proposed self-supervised method accelerates inference efficiently while maintaining or improving generation quality, offering a flexible and model-agnostic enhancement to diffusion sampling.

**Strengths:**

The literature review seems to be comprehensive.

**Weaknesses:**

* The writing and the logic of this paper can be improved greatly. For example, the introduction is almost as the same length as the abstract. With such a short paragraph, the authors do not make it clear about the background and the motivation of their proposed methods. Overall, this paper gives me a feeling that the authors are a little bit rush to catch the ICLR due, and does not finish the paper.

* The novelty is limited. This paper is not the first paper to introduce reinforcement learning to learn a noise schedule in diffusion model sampling process. The authors should discuss their difference, and novelty compared the previous methods.

* The experiment is also not comprehensive enough. There are not compared methods in the experiment. The experiments are also conducted on few small-scale datasets. More evaluations and ablation study are needed to justify the advantage of the proposed method. Also, the experiment analysis is very limited.

* Typos: There are lots of typos in this paper, and the authors should fix them carefully. For example, in Line 121, “Or” after the comma should be lower bracketed.

**Questions:**

* I believe the quadric sampling performs the best in CIFAR10 and CelebA-HQ. Why do the authors not include the results of using quadric sampling?

* I am wondering if there is a constraint on the output of the policy $\pi_\theta$. As the experiment are conducted with the same time steps (like in 10 steps). How to ensure the proposed method can just complete the sampling process in 10 steps?

---

> ### Author Response · Authors · 2025-11-21
>
> We would like to address the concerns raised by the reviewer below:
>
> 1. Regarding novelty limitation, the reviewer claims that we are not the first paper to introduce reinforcement learning to learn a noise schedule in diffusion model sampling. In our paper, we mention a similar existing work (Learning to Schedule in Diffusion Probabilistic Models by Wang et al.  https://dl.acm.org/doi/10.1145/3580305.3599412 ) which uses paired examples by running a diffusion model for large number of steps. Our novelty comes from the self-supervised reward, which unlike the method mentioned, does not need to compute expensive pairings. We would appreciate it if the reviewer could provide references if there are other works that he had in mind when mentioning the limits of our novelty, or clarify that which method he was talking about.
>
> 2. As mentioned in Section 4.1, our policy’s action space is [0, 1], meaning it acts unconstrained. It is actually the core of the method that our policy is allowed to terminate the diffusion process before expanding the entirety of its budget, as shown in figure 2.
>
> 3. The reviewer states: "I believe the quadric sampling performs the best in CIFAR10 and CelebA-HQ.".  We are not aware of such claims. Can the reviewer provide references ?

---

### Official Review · Reviewer_DpCq · 2025-10-27

**Soundness:** 2
**Presentation:** 2
**Contribution:** 2
**Rating:** 2
**Confidence:** 4

**Summary:**

The paper proposes a reinforcement learning (RL) framework to accelerate diffusion model sampling without retraining the base model. Traditional diffusion models generate high-quality images but are slow at inference because of iterative denoising. The authors treat the denoising process as a sequential decision-making task, where an RL agent learns an optimal noise schedule to balance speed and sample quality.

**Strengths:**

1. Positioning RL as a learned scheduler for diffusion sampling is a forward-looking direction with clear potential: it decouples the sampler from the base model, enables adaptive step allocation under budgets, and could generalize beyond image generation to other iterative inference/integration tasks.

2. Self-Supervised Design: Avoids dependence on paired data or teacher models, reducing computational cost and improving scalability.

3. General Applicability: Works as a plug-in sampler for any pretrained diffusion model without retraining.

**Weaknesses:**

1. Limited Evaluation: Only small-scale datasets (CIFAR-10, CelebA-HQ) — no large or diverse benchmarks to test generalization.

2. Unclear Comparisons: Some baselines (e.g., DPM++) perform poorly but reasons are not analyzed; RL results on second-order solvers are weaker.

3. Missing Ablation Studies: No detailed analysis on hyperparameters, reward sensitivity, or computational overhead.

4. Paper organization & contribution strength. The manuscript’s sectioning and exposition require substantial tightening (clearer motivation, positioning vs. prior work). As presented, the contribution appears incremental with limited novelty and empirical breadth, which falls short of the ICLR acceptance bar.

**Questions:**

1. High-resolution & scale. Modern diffusion systems often operate at 512×512 or 1024×1024. How does the proposed RL scheduler perform at higher resolutions and on large-scale datasets (e.g., ImageNet-256/512)? Please report both quality (FID/KID) and wall-clock latency.

2. Real speedup under matched quality. Beyond NFEs, what is the end-to-end latency per image (and throughput) on a standard GPU compared to strong samplers when targeting the same quality? Include hardware, batch size, and confidence intervals.

3. Practicality: training cost and transfer. What compute/time is required to train the RL policy per model/dataset, and does the policy transfer across architectures or noise parameterizations? If retraining is needed, what’s the break-even point where the amortized gains outweigh the training overhead?

---

> ### Author Response · Authors · 2025-11-21
>
> We thank the reviewer for their feedback and would like to address each concern and question below.
>
> 1. We have not experimented with higher-dimensional datasets and agree that it would be interesting next steps
>
> 2. Our policy inference cost is orders of magnitude faster than the diffusion models themselves. For a typical single diffusion step, our policy represents around 5% of the inference time. We agree that NFE vs FLOPs/wallclock time would be a needed evaluation metric
>
> 3. The compute / time required to train the RL policy depends on the dataset as well as the budget, however it is relatively quick. For example, CIFAR-10 with a budget of 10-NFE on an NVIDIA L40s trains in under 3 hours. We have not experimented with transfer learning across datasets which would drastically increase amortized gains, which could be a valuable extension.

---

### Official Review · Reviewer_Dcu1 · 2025-10-30

**Soundness:** 2
**Presentation:** 2
**Contribution:** 2
**Rating:** 2
**Confidence:** 5

**Summary:**

The paper introduces a reinforcement learning (RL)-based scheduler to accelerate diffusion model sampling. Unlike prior work that retrains or distills diffusion models, this approach learns an optimal denoising schedule via RL without retraining or requiring paired supervision. The method frames diffusion inference as an MDP, where each denoising step is an action, and a Gaussian policy learns to decide adaptive noise updates. A self-supervised reward is designed by measuring the maximum similarity between generated samples and real data features (using Inception-v3 embeddings). The approach is evaluated on CIFAR-10 and CelebA-HQ using DDIM and IADB diffusion models, achieving better FID scores with fewer function evaluations (NFEs).

**Strengths:**

1. The framing of diffusion sampling as a sequential decision-making problem is elegant and consistent with the Markovian structure of denoising.

2. The RL component (PPO-based scheduler) is model-agnostic—it does not modify or retrain the base diffusion network, enabling plug-and-play acceleration across architectures (DDIM, IADB).

3. Using feature-space similarity as a reward proxy allows the method to circumvent supervised alignment and label requirements.

**Weaknesses:**

1. A significant concern is that the proposed “self-supervised” reward actually depends on an Inception-v3 feature encoder pretrained on ImageNet, which likely leaks high-level real-image information into the training process. This coupling may bias the RL scheduler toward reproducing specific training-set patterns and artificially improve FID, since both the reward and evaluation metrics are derived from the same embedding space. Consequently, the reported performance gains might not reflect genuine improvements in sample quality or generalization. Future work should decouple the reward signal from FID features or use domain-specific encoders to ensure fairness and generalizability.

2. Despite claiming theoretical soundness, the paper lacks a rigorous proof of convergence or guarantee of policy optimality. The “theoretical grounding” in Appendix A only discusses FID variance bounds, not RL convergence or sampling correctness.

3. The writing of the paper and the figure 1 seem inconsistent and is confusing in terms of whether the RL training leverages sparse vs dense reward. If the training requires sparse reward design, the PPO training will be unstable and requires more explorations and data collection for the RL algorithm to converge. The paper would benefit from a dense-reward design so that PPO works better though it comes with more computational cost.

4. Key learning-based baselines [1,2] are missing. Without these, it’s difficult to attribute performance gains to the proposed self-supervised RL design.

5. No ablation studies are presented to isolate the contribution of self-supervision vs. policy learning.

[1] Lu, Cheng, et al. "Dpm-solver++: Fast solver for guided sampling of diffusion probabilistic models." Machine Intelligence Research (2025): 1-22.

[2] Xue, Shuchen, et al. "Sa-solver: Stochastic adams solver for fast sampling of diffusion models." Advances in Neural Information Processing Systems 36 (2023): 77632-77674.

**Questions:**

1. You claim that the method is self-supervised, yet the reward relies on features extracted from a pretrained Inception-v3 model. How do you reconcile this with the self-supervised claim, given that Inception features encode semantic information learned from large-scale labeled datasets?

2. The reward is sparse (terminal-only) and non-differentiable. How stable is PPO training under this setup? Did you attempt intermediate-step rewards or reward shaping to improve credit assignment?

3. PPO is known to be sensitive to hyperparameters. Could you report variance over multiple seeds or provide details on convergence behavior?

4. How does your policy network scale with higher-dimensional datasets or larger image resolutions (e.g., 512×512 or beyond CIFAR-10/CelebA-HQ)?

---

> ### Author Response · Authors · 2025-11-21
>
> We thank the reviewer for their feedback and would like to address each concern and question below.
>
> 1. We claim that we are self-supervised, because we do not provide target pairs or ground truths to our policy for denoising. While Inception-v3 itself is pre-trained using supervised learning on ImageNet, our pipeline does not involve any human-labeled or paired data. Even if embeddings themselves are trained via classical supervised learning, a pipeline using these embeddings with unlabeled data such as ours can still be labeled as self-supervised.
>
> 2. We did experiment with providing rewards at every step rather than only terminal, however we observed that this produces unwanted behavior. Because we guarantee episode termination, our training is relatively stable, since each episode will always end with a good signal to the PPO learner.
>
> 3. While we do not provide theoretical convergence guarantees of policy optimality, this is a broader limitation shared across most deep RL research with non linear function approximation, especially with neural environments such as in our setting. We agree that empirical results with multiple runs over different seeds might provide sufficient details on convergence behavior.
>
> 4. We have not experimented with higher-dimensional datasets and agree that it would be interesting next steps

---

### Official Review · Reviewer_oRLs · 2025-10-31

**Soundness:** 3
**Presentation:** 3
**Contribution:** 4
**Rating:** 6
**Confidence:** 4

**Summary:**

This paper introduces a reinforcement learning (RL) approach to accelerate diffusion model sampling by learning an adaptive noise schedule policy. The method frames the diffusion sampling process as an RL episode where each denoising step corresponds to an environment step. The key innovation is a self-supervised reward based on maximum pairwise similarity between generated samples and real data in a pretrained feature space (Inception-v3), eliminating the need for teacher models or paired data. Experiments on CIFAR-10 and CelebA-HQ with DDIM and IADB models show that the learned scheduler achieves competitive or better FID scores compared to uniform and cosine schedules, particularly in low-compute regimes (10-30 NFEs).

**Strengths:**

- **Self-Supervised Reward:** Uses feature similarity instead of supervised signals, enabling broader applicability.
- **Model-Agnostic:** Works with any pretrained diffusion model (DDIM, IADB) and sampler order.
- **Efficiency:** Achieves better FID with fewer NFEs, especially in resource-constrained settings.
- **Theoretical Insight:** Appendix A provides a bound on FID estimation error due to finite sampling.

**Weaknesses:**

- **Inconsistent Second-Order Results:** The method underperforms on second-order solvers for CIFAR-10, raising questions about its generalizability.
- **Limited Reward Analysis:** The choice of Pearson correlation is not thoroughly justified; other metrics (e.g., cosine similarity) are not compared.
- **Ablation Studies Missing:** No analysis of the impact of batch size, feature extractor choice, or policy architecture.
- **Narrow Evaluation:** Only FID is used; no diversity metrics (e.g., Precision/Recall) or human evaluation.

**Questions:**

1. **Why does the method underperform on second-order solvers for CIFAR-10?** Is this due to the reward function or the complexity of the policy?
2. **Have you experimented with other similarity metrics (e.g., cosine similarity) or feature extractors?** An ablation would strengthen the reward design.
3. **How does the batch size of real samples \(D\) affect performance?** Is there a trade-off between reward quality and computational cost?
4. **Can the policy generalize to NFE budgets not seen during training?**
5. **Have you considered using dense rewards (e.g., intermediate similarity scores) to guide learning?**

---

> ### Author Response · Authors · 2025-11-21
>
> We thank the reviewer for their feedback and would like to address each concern and question below.
>
> 1. We are not quite sure why the second order solver underperforms for CIFAR-10 especially in low NFE regimen. Future work should investigate this behavior.
>
> 2. We have experimented with many different similarity metrics (cosine similarity, pearson correlation, L2 distance, SSIM). Both cosine similarity and pearson correlation behaved equally well. However a thorough ablation study would strengthen the claim
>
> 3. Increasing the batch size of real samples D would lead to smoother rewards. At the limit, if we could set D = to the entire dataset, we would have no stochasticity in the reward and smoother learning, although this would not be computationally feasible. Our lower bound analysis of FID due to finite sampling aims to provide more theoretical intuition behind the choice of D.
>
> 4. We have not experimented to NFE budgets not seen during training, and we note that we train a separate policy per NFE budget.
>
> 5. We did experiment with providing rewards at every step rather than only terminal, however we observed that this produces unwanted behavior.

---

> > ### Comment · Reviewer_oRLs · 2025-11-26
> >
> > We thank the authors for their thoughtful response to our initial review. We appreciate the clarifications provided and are pleased to see that the authors have acknowledged several of the limitations we raised. Below, we offer a follow-up comment to further strengthen the dialogue and potential revisions:
> >
> > 1. **Regarding the second-order solver performance on CIFAR-10**:
> >    We appreciate the authors' honesty in noting the inconsistent performance and their commitment to future investigation. We encourage the authors to include a brief discussion of this phenomenon in the final version—even if a full explanation remains open—as it would provide valuable insight for the community.
> >
> > 2. **Similarity metrics and reward design**:
> >    It is reassuring to learn that both Pearson correlation and cosine similarity performed well. We strongly recommend including a small ablation study or at least a summary of these experiments in the appendix, as this would substantiate the reward design and enhance reproducibility.
> >
> > 3. **Batch size and reward stability**:
> >    The authors' reference to the theoretical bound in Appendix A is a strong point. We suggest explicitly connecting this analysis to the practical choice of batch size in the main text, which would help readers understand the trade-off between computational cost and reward reliability.
> >
> > 4. **Generalization to unseen NFE budgets**:
> >    While training separate policies per budget is a valid approach, we encourage the authors to discuss the limitations of this design in the paper, and if possible, to explore or speculate on methods for generalizing across budgets in future work.
> >
> > 5. **Dense vs. sparse rewards**:
> >    The authors’ experimental finding that dense rewards led to undesirable behavior is an interesting result in itself. We recommend briefly describing this outcome in the paper, as it could help other researchers avoid similar pitfalls.
> >
> > Overall, we believe that addressing these points—even partially—would significantly improve the paper's clarity, rigor, and impact. We look forward to seeing a revised version that incorporates these suggestions.

---

### Meta-Review · Area_Chair_ZFxn · 2025-12-28

**Summary:**

This paper proposes to learn a denosing scheduler for diffusion models using reinforcement learning, with the maximum feature similarity between the generated image and real samples as the reward signal. The paper received four reviews, with the majority of them on the negative side.

Reviewers commented positively on the formulation of sequential decision-making, the model-agnostic nature of the proposed method, and self-supervised learning using feature similarity.

Reviewers raised significant concerns about the writing quality of the paper, the limited novelty, the insufficient or inconsistent experiments, and the missing ablation studies.

Overall, according to reviewers' recommendations, the weaknesses outweigh the strengths.

**Reviewer Concerns:**

(Weaknesses are indexed using reviewers' original ordering)

For reviewer oRLs, W1-3 have been partially addressed and acknowledged by the reviewer. W4 has not been addressed.

For reviewer Dcu1, W1 has been addressed well. W2-5 are still outstanding.

For reviewer DpCq, W1-4 are all still outstanding.

For reviewer som6, W2 has been partially addressed. W1, W3, and W4 are outstanding.

**Reviewer Scores:**

For reviewer oRLs, the reviewer has responded, and it is unlikely to increase the score further.

For reviewer Dcu1, the score is unlikely to be increased.

For reviewer DpCq, the score is unlikely to be increased.

For reviewer som6, the score is unlikely to be increased.

---

### Decision · Program_Chairs · 2026-01-26

Reject